**DOI: 10.1038/ncomms13816**　**OPEN**

# PAXX promotes KU accumulation at DNA breaks and is essential for end-joining in XLF-deficient mice

Xiangyu Liu[1], Zhengping Shao[1], Wenxia Jiang[1], Brian J. Lee[1] & Shan Zha[1,2]

Non-homologous end-joining (NHEJ) is the most prominent DNA double strand break (DSB) repair pathway in mammalian cells. PAXX is the newest NHEJ factor, which shares structural similarity with known NHEJ factors—XRCC4 and XLF. Here we report that PAXX is dispensable for physiological NHEJ in otherwise wild-type mice. Yet $Paxx^{-/-}$ mice require XLF and $Xlf^{-/-}$ mice require PAXX for end-ligation. As such, $Xlf^{-/-}Paxx^{-/-}$ mice display severe genomic instability and neuronal apoptosis, which eventually lead to embryonic lethality. Despite their structural similarities, only $Xlf^{-/-}$ cells, but not $Paxx^{-/-}$ cells require ATM/DNA-PK kinase activity for end-ligation. Mechanistically, PAXX promotes the accumulation of KU at DSBs, while XLF enhances LIG4 recruitment without affecting KU dynamics at DNA breaks *in vivo*. Together these findings identify the molecular functions of PAXX in KU accumulation at DNA ends and reveal distinct, yet critically complementary functions of PAXX and XLF during NHEJ.

[1] Department of Pathology and Cell Biology, College of Physicians and Surgeons, Institute for Cancer Genetics, Columbia University, 1130 Saint Nicholas Avenue, Room 501, New York City, New York 10032, USA. [2] Division of Pediatric Oncology, Hematology and Stem Cell Transplantation, Department of Pediatrics, College of Physicians & Surgeons, Columbia University, 1130 Saint Nicholas Avenue, Room 501, New York City, New York 10032, USA. Correspondence and requests for materials should be addressed to S.Z. (email: sz2296@cumc.columbia.edu).

DNA double strand breaks (DSBs) are the most severe form of DNA damages. In mammalian cells, most DSBs are repaired by the non-homologous end-joining (NHEJ) pathway, which directly ligates two DNA ends together. NHEJ is not only critical for general DNA repair, but also exclusively required for lymphocyte development and the viability of post-mitotic neurons[1]. Accordingly, human patients and mouse models with defects in any of the seven well-characterized NHEJ factors develop severe primary immunodeficiency and variable degrees of microcephaly[1,2]. Among these factors, KU70 and KU80 (KU86 in human) form a heterodimer (referred to as KU) that binds and protects DNA ends from nucleases and recruits the DNA-dependent protein kinase catalytic subunit (DNA-PKcs). DNA-PKcs in turn recruits and activates the Artemis endonuclease for end-processing (for example, hairpin opening). Finally, DNA Ligase 4 (Lig4), in complex with XRCC4 and XLF (also called Cernunnos or NHEJ1), ligates the ends together. The five 'core' NHEJ factors, KU70, KU80, XRCC4, Lig4 and XLF, are essential for end-ligation and conserved in all eukaryotes. In contrast, complete loss of DNA-PKcs or Artemis, both of which are only found in vertebrates, abolishes end-processing (that is, hairpin opening), but not end-ligation. We recently reported that expression of a kinase-dead DNA-PKcs protein blocks end-ligation, suggesting that DNA-PKcs coordinates end-processing and end-ligation during NHEJ through auto-phosphorylation[3].

Normal lymphocyte development requires the assembly of functional antigen receptor genes from germline variable (V), diversity (D) and joining (J) gene segments through V(D)J recombination. V(D)J recombination is initiated by products of the recombination activating gene (RAG), which cleaves germline antigen receptor genes to generate a pair of blunt signalling ends (SEs) and a pair of hairpin coding ends (CEs)[4]. The two SEs are precisely joined by the core NHEJ factors to form a signalling joint (SJ), while the two hairpin CEs are first opened by DNA-PKcs and Artemis before being joined by the core NHEJ factors to generate a coding joint (CJ). The unique structures of CEs and SEs and the exclusive requirement for NHEJ during V(D)J recombination provide a sensitive and well-characterized physiological system to evaluate end-ligation and end-processing functions of the NHEJ factors. While loss of the conserved core NHEJ factor abrogates both CJs and SJs, complete loss of DNA-PKcs or Artemis only abrogates CJ formation. CJs comprise part of the exons encoding the antigen receptor genes and are thus required for lymphocyte development. As such, loss of all previously characterized NHEJ factors, except XLF (see below), abrogates lymphocyte development at the progenitor stage, leading to $T^- B^-$ severe combined immunodeficiency (SCID) in patients and animal models. RAG holds the DNA ends generated during V(D)J recombination in a post-synaptic complex, which promotes end-ligation in the absence of XLF[5]. In peripheral lymphoid organs (for example, the spleen), naive B cells further modify the constant region of the immunoglobulin heavy chain (IgH) through class switch recombination (CSR) to achieve different antibody effector functions. CSR is initiated by activation-induced deaminase. The DSB intermediates generated during CSR, which do not harbour hairpins, are joined together primarily by core NHEJ factors to complete CSR. Thus, loss of any core NHEJ factor, including XLF, leads to frequent IgH chromosomal breaks and reduces CSR efficiency by 50–75% (refs 6–8). In addition, post-mitotic neurons require NHEJ for survival. Complete loss of core NHEJ factors causes widespread neuronal apoptosis and eventually late embryonic lethality in Lig4- or XRCC4-deficient mice[1,3,9–11]. Efficient and accurate NHEJ on chromatinized DNA also requires ataxia-telangiectasia mutated (ATM) kinase and ATM-mediated DNA damage responses[6]. While not essential for V(D)J recombination, loss of ATM or its substrates (for example, H2AX and 53BP1) abrogates chromosomal V(D)J recombination in XLF-deficient mice and cells, underscoring the intricate interaction between DNA damage response and core NHEJ factors[12–15].

Paralog of XRCC4 and XLF (PAXX, also called C9ORF142 or XLS) was proposed as a NHEJ factor based on its structural similarity with XRCC4 and XLF[16–18]. Since patients or animal models with defects in PAXX are not yet found, the physiological function of PAXX remains largely unknown. XRCC4, XLF and PAXX all have an N-terminal globular head domain followed by a C-terminal coil-coiled stalk, and each forms stable homodimers via their respective coil-coiled stalks[19,20]. XRCC4 deficiency phenocopies Lig4 deficiency, likely because the stalk of the XRCC4 homodimer binds and stabilizes Lig4 protein. In contrast, the coiled-coil stalks of XLF and PAXX are much shorter and do not bind Lig4 directly[16,17,21,22]. While not absolutely required for NHEJ, XLF dimers promote end-ligation by forming high-order helical filaments with XRCC4 dimers through direct interactions between their respective head domains[7,12,23,24]. PAXX does not directly interact with either XLF or XRCC4. Instead, PAXX binds KU through a conserved C-terminal region[16–18]. A PAXX mutant that cannot bind to KU fails to rescue the severe IR sensitivity in human cells[16,18]. Notably, co-deletion of PAXX partially rescues the severe IR sensitivity of XRCC4-knockout DT40 cells[17], but accentuates the zeocin sensitivity of XLF-deficient HCT116 cells[24]. The exact role of PAXX in NHEJ and DSB repair is yet to be demonstrated.

To elucidate the functions of PAXX in NHEJ and determine the physiological function of PAXX in vivo, we generated PAXX-deficient mice. Interestingly, although not required for NHEJ in otherwise wild-type mice, PAXX is essential for both chromosomal and extra-chromosomal end-ligation in Xlf-deficient animals. Our data further suggest that PAXX and XLF support NHEJ through distinct mechanisms, with PAXX facilitating the accumulation of KU at DNA ends and XLF enhancing the recruitment of Lig4 to achieve efficient end-ligation together.

## Results

**Generation of PAXX-deficient mice.** To determine the physiological functions of PAXX, we generated Paxx knockout mice ($Paxx^{-/-}$) by a gene targeting strategy in which all coding exons of the Paxx gene and part of the non-coding exon 1 were replaced by a Neomycin resistant (NeoR) cassette flanked by frt sequences (Fig. 1a). Correct targeting, which removes an EcoRV site within the Paxx gene, was confirmed by Southern blotting analyses (Fig. 1b). Eight independently targeted embryonic stem (ES) cell clones (in 129/sv background) were obtained and two were injected for germline transmission. The resulting chimeras were bred with $ROSA26^{FLIP/FLIP}$ mice expressing FLIPase constitutively[25] (Jackson Laboratory, Stock No. 003946) to remove the NeoR cassette and generate $Paxx^{+/-}$ mice. Intercrosses between $Paxx^{+/-}$ mice produced $Paxx^{-/-}$ pups at the expected Mendelian ratio (Fig. 1c). Southern blotting confirmed complete deletion of the Paxx gene in $Paxx^{-/-}$ mice and western blotting validated the absence of PAXX protein in $Paxx^{-/-}$ murine embryonic fibroblasts (MEFs) (Fig. 1d,e). Both male and female $Paxx^{-/-}$ mice are of normal size (Fig. 1f), and spontaneous tumour development has not yet been observed. Given the embryonic lethality and severe growth retardation of XRCC4- and KU-deficient mice, these data suggest that PAXX is not essential for end-ligation in mice.

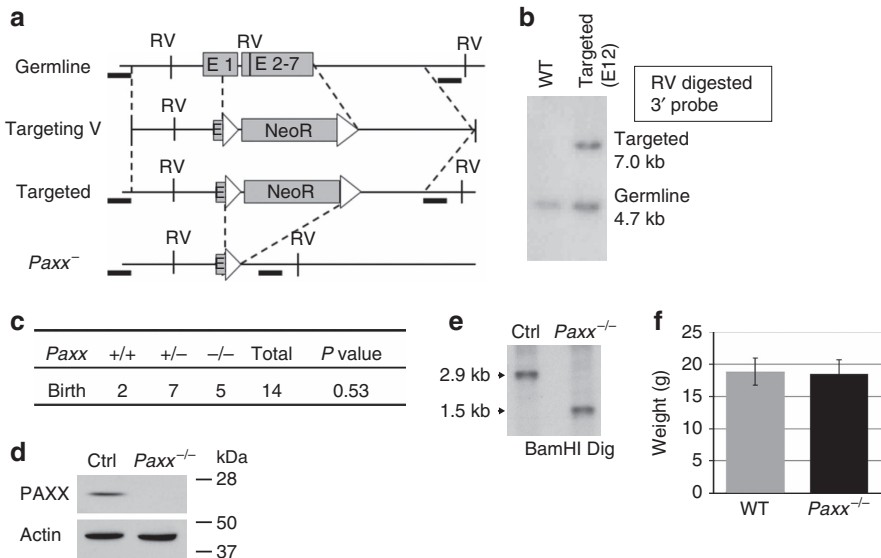

**Figure 1 | Generation of $Paxx^{-/-}$ mice.** (**a**) The schematic diagram represents the murine *Paxx* locus (top), targeting vector (2nd row), targeted allele (3rd row) and the neo-deleted allele ($Paxx^-$, bottom). The 5′ and 3′ probe are marked as thick black lines. The exons and frt sites are shown as solid boxes and open triangles, respectively. Restriction site designation: RV = EcoRV. The map is not drawn to scale. (**b**) Southern blot analyses of EcoRV-digested DNA from WT and *Paxx* + /Targeted ES cells, blotted with the 3′ probe. (**c**) The number of live-birth mice obtained from intercrossing $Paxx^{+/-}$ mice. The *P* value was calculated with the chi-square test. (**d**) Western blot for PAXX in primary murine embryonic fibroblasts derived from E14.5 WT or $Paxx^{-/-}$ embryos. (**e**) Southern blot analyses of BamHI-digested DNA from $Paxx^{+/+}$ and $Paxx^{-/-}$ mice tissue DNA and probed with mPaxx probe diagrammed in Fig. 5a. (**f**) The total body weight of $Paxx^{+/+}$ and $Paxx^{-/-}$ littermates at ∼50 days of age. The data represent the average and s.d. of more than four mice of each genotype.

**Lymphocyte development is normal in $Paxx^{-/-}$ mice.** Since $Xlf^{-/-}$ mice develop normally, but have reduced lymphocyte cellularity and CSR defects[7], we next analysed $Paxx^{-/-}$ mice for lymphocyte development. The weights and total cellularity of the thymus and spleens of 7–8-week-old $Paxx^{-/-}$ mice (4 male, 6 female) were indistinguishable from those of $Paxx^{+/+}$ (4 male, 3 female) littermates (Fig. 2a and Supplementary Fig. 1A). Fluorescence activated cell sorting (FACS) analyses showed that the frequencies of immature pro-B (CD43 + B220 + IgM − ), pre-B (CD43 − B220 + IgM − ), newly generated naive B (IgM + B220^low) and re-circulating B (IgM + B220^hi) cells in bone marrow from $Paxx^{-/-}$ mice were also comparable to those of $Paxx^{+/+}$ mice (Fig. 2b and Supplementary Fig. 1C). Successful V(D)J recombination at the IgH locus is required for the transition from pro-B to pre-B cells. The ratio of bone marrow-derived pre-B/pro-B cells was the same in both $Paxx^{-/-}$ mice and their $Paxx^{+/+}$ littermates (Fig. 2b). Likewise, the number and frequency of T-cell progenitors and mature T cells are also indistinguishable in $Paxx^{-/-}$ and $Paxx^{+/+}$ mice (Fig. 2c and Supplementary Fig. 1B). Sequential rearrangements of the TCRα locus in CD4 + CD8 + double positive (DP) thymocytes are coupled with both positive and negative selections, creating a stressful situation that reveals minor V(D)J recombination defects in ATM or 53BP1-deficient cells previously, indicated by reduced surface expression of TCRβ and its co-receptor CD3 in the DP cells[26,27] (Supplementary Fig. 1D). Nevertheless, surface expression of TCRβ/CD3 in DP cells was not affected by PAXX deficiency (Fig. 2c). Consistent with normal lymphocyte development, endogenous V(D)J recombination junctions in $Paxx^{-/-}$ and $Paxx^{+/+}$ mice are also indistinguishable (Supplementary Table 1). Moreover, CSR is not affected by *Paxx* deficiency. *In vitro* stimulation with bacterial lipopolysaccharide (LPS) and interleukin 4 (IL-4) induced robust CSR to IgG1, and expression of surface IgG1 in ∼30% of $Paxx^{+/+}$ as well as $Paxx^{-/-}$ B cells in 4 days (Figs 2d,e and Supplementary Fig. 2A), while only ∼10% of $Xlf^{-/-}$ B cells express IgG1 at Day 4, consistent with 50–75% reduction in CSR[7]. Likewise, proliferation of *in vitro*-stimulated $Paxx^{-/-}$ and $Paxx^{+/+}$ B cells is also similar (Supplementary Fig. 2B). Together, these results indicate that PAXX, unlike XLF and other NHEJ factors, is not required for either V(D)J recombination or CSR, two physiological gene rearrangements mediated by NHEJ.

**$Xlf^{-/-}Paxx^{-/-}$ mice die during embryonic development.** Previous studies by us and others suggest that the apparently efficient chromosomal end-joining in XLF-deficient cells is quite vulnerable and requires both ATM and DNA-PK kinase activities and their substrates (for example, 53BP1 and H2AX)[5,12–14,24,28]. Given the structural similarity between PAXX and XLF, we asked whether PAXX is necessary for end-joining in $Xlf^{-/-}$ mice. Intercrossing double-heterozygous $Xlf^{+/-}Paxx^{+/-}$ mice yielded $Xlf^{-/-}Paxx^{+/-}$ and $Xlf^{+/-}Paxx^{-/-}$ pups at the expected Mendelian ratios, similar to those of $Xlf^{-/-}Paxx^{+/+}$ and $Xlf^{+/+}Paxx^{-/-}$ pups (Fig. 3a). However, double-homozygous $Xlf^{-/-}Paxx^{-/-}$ progeny were not obtained, suggesting embryonic lethality (Fig. 3a). Embryonic lethality caused by Lig4 or Xrcc4 deficiency occurs shortly after embryonic day 15.5 (E15.5) with severe neuronal apoptosis[3,9–11]. Notably at E14.5, $Xlf^{-/-}Paxx^{-/-}$ embryos, which were slightly smaller in size than their littermates, were found at the expected Mendelian ratios (Fig. 3b). Neuronal apoptosis, as indicated by condensed nuclei and positive staining for cleaved (activated) Caspase 3, was rare in $Xlf^{-/-}$ or $Paxx^{-/-}$ embryos and increased >40-fold in $Xlf^{-/-}Paxx^{-/-}$ embryonic brains, to a level comparable to end-ligation defective $Xrcc4^{-/-}$ or $DNA-PKcs^{KD/KD}$ embryos (Fig. 3c,d)[3,11]. Moreover, apoptotic inclusions were most prominent in the post-mitotic intermediate zone, but not the proliferating ventricular zone,

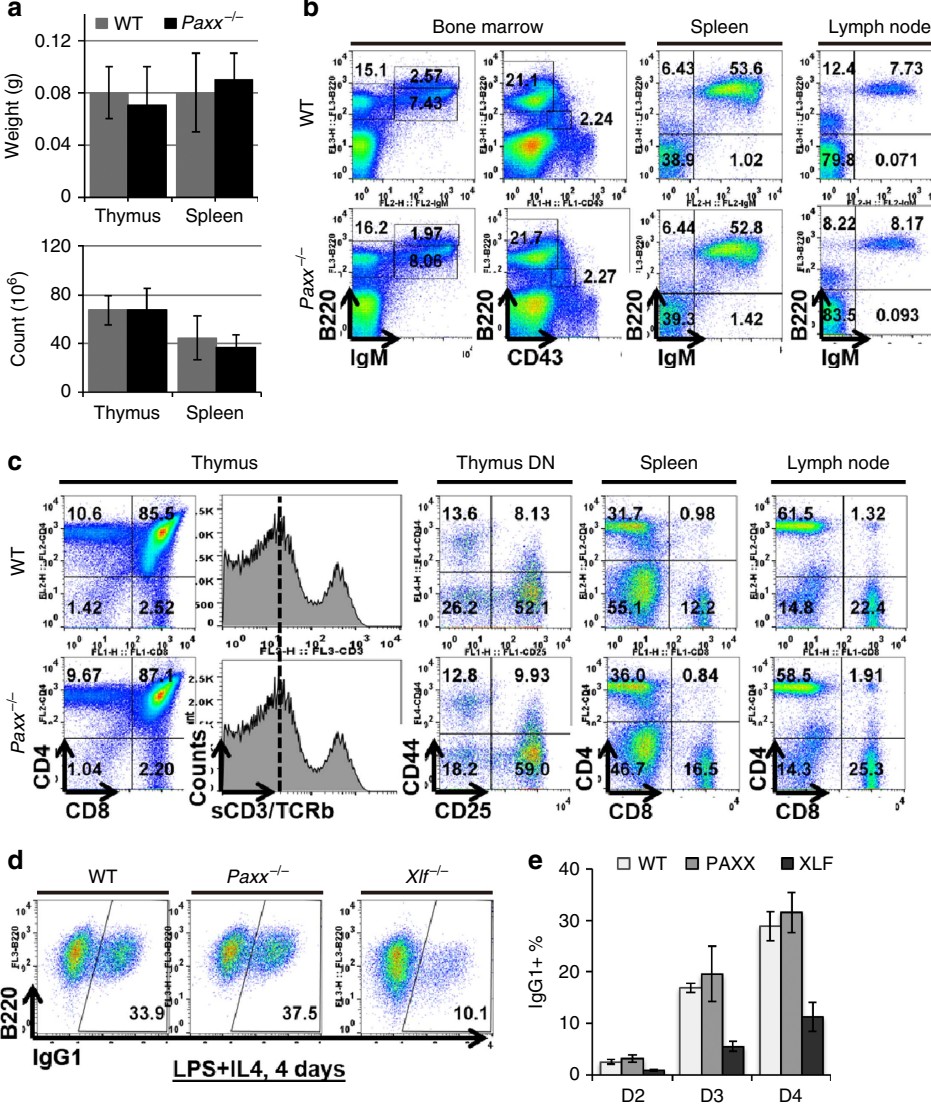

**Figure 2 | Lymphocyte development in *Paxx*$^{-/-}$ mice.** (**a**) Weight and total cell count of thymus and spleen from *Paxx*$^{+/+}$ and *Paxx*$^{-/-}$ littermates at ~50 days of age (7–8 weeks). The data represent the average and s.d. of >3 mice of each genotype. (**b,c**) Representative flow cytometry analyses of 7–8-week-old *Paxx*$^{+/+}$ and *Paxx*$^{-/-}$ mice. (**d**) Representative flow cytometry analysis of IL-4/LPS stimulated B cells (CD43$^-$ splenocytes) from *Paxx*$^{+/+}$, *Paxx*$^{-/-}$ and *Xlf*$^{-/-}$ mice. (**e**) Statistical analyses of the class-switching results in **d**. Three biological repeats were performed for each genotype. The bar graphs represent the average and s.e.

of *Xlf*$^{-/-}$ *Paxx*$^{-/-}$ embryos, in a manner reminiscent of *Xrcc4*$^{-/-}$ embryos[29]. Together, these findings reveal a critical role of PAXX in the survival of post-mitotic neurons and embryonic development in mice lacking XLF.

**Severe genomic instability in *Xlf*$^{-/-}$ *Paxx*$^{-/-}$ cells.** To determine whether DNA repair defects and genomic instability contribute to the embryonic lethality of *Xlf/Paxx* double-deficient mice, we generated MEFs from E14.5 *Xlf*$^{-/-}$ *Paxx*$^{-/-}$ and control (wild type (WT) and *Xlf-* or *Paxx* single deficient) embryos. Consistent with our previous study[30] and the normal development of *Paxx*$^{-/-}$ mice, the *Paxx*$^{-/-}$ and *Xlf*$^{-/-}$ primary MEFs proliferated at rates comparable to that of WT MEFs. In contrast, *Xlf*$^{-/-}$ *Paxx*$^{-/-}$ MEFs displayed severe proliferation defects and failed to thrive at early passages (Fig. 4a). Correspondingly, cell cycle analyses revealed ~25% reduction (from 25 to ~18%) in the percentage of S phase (BrdU+) cells

from *Xlf*$^{-/-}$ *Paxx*$^{-/-}$ primary MEFs (Supplementary Fig. 3A). Despite normal proliferation, *Paxx*$^{-/-}$ MEFs were moderately, yet significantly, more sensitive to IR and Etoposide than *Paxx*$^{+/+}$ cells, while less sensitive than *Xlf*$^{-/-}$ MEFs (Fig. 4b and Supplementary Fig. 3B). Meanwhile, *Xlf*$^{-/-}$ *Paxx*$^{-/-}$ MEFs were highly sensitive to IR and Etoposide, consistent with their severe proliferation defects (Fig. 4b). In addition, *Paxx*$^{-/-}$ and *Xlf*$^{-/-}$ primary MEFs are very moderately sensitive to hydroxyurea (HU) at the highest dose, while *Xlf*$^{-/-}$ *Paxx*$^{-/-}$ primary MEFs are not apparently sensitive to HU (Supplementary Fig. 3C). Since HU primarily targets replicating cells, the lack of HU sensitivity might be due to the lower frequency of S phase cells among *Xlf*$^{-/-}$ *Paxx*$^{-/-}$ primary MEFs (Supplementary Fig. 3A). Cytogenetic analyses of spontaneous genomic instability using telomere-specific fluorescence *in situ* hybridization (T-FISH) revealed that 18.9 ± 2.6% *Xlf*$^{-/-}$ *Paxx*$^{-/-}$ primary MEFs had at least one cytogenetic abnormality, in comparison with

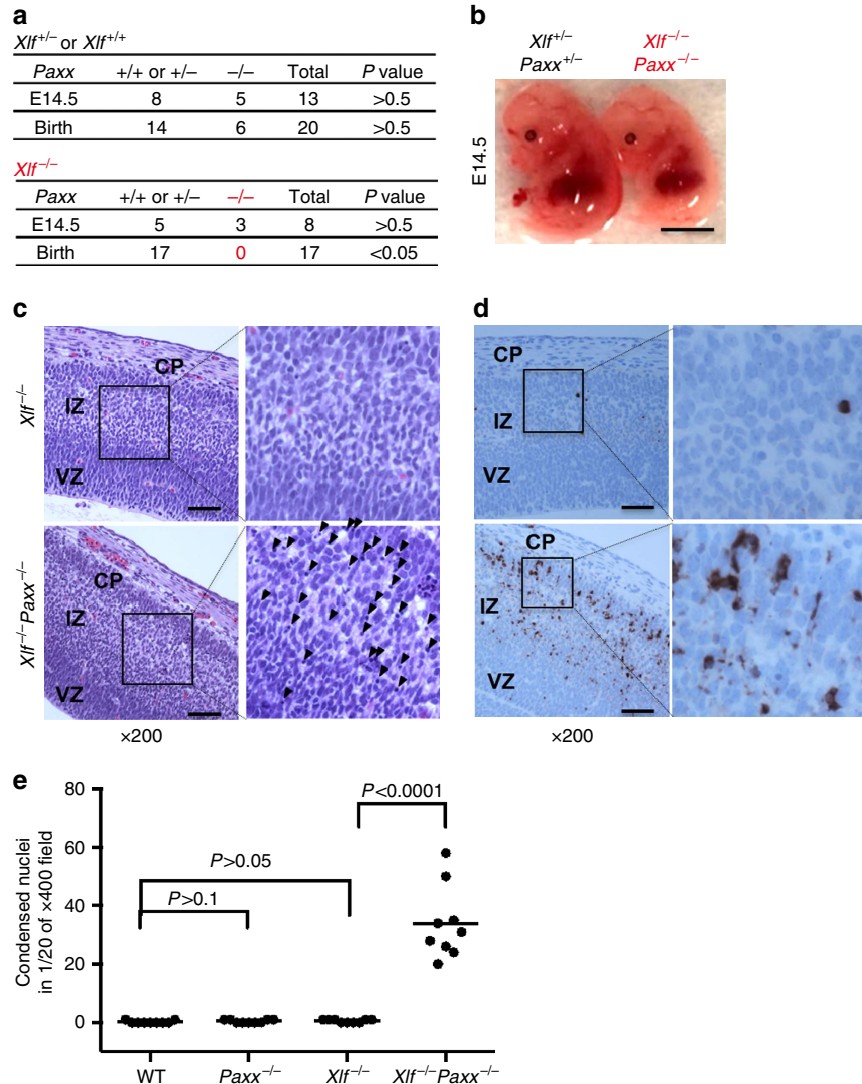

**Figure 3 | Embryonic lethality and severe neuronal apoptosis in $Xlf^{-/-}Paxx^{-/-}$ mice.** (a) Live birth and E14.5 embryos obtained from intercross between Xlf and Paxx-deficient mice. (Upper table) All E14.5 embryo or birth with $Xlf^{+/+}$ or $Xlf^{+/-}$ genotypes and various Paxx genotypes. (Lower table) All E14.5 embryos or birth with $Xlf^{-/-}$ genotypes. $Xlf^{-/-}Paxx^{-/-}$ were not found at birth and found at expected ratio at E14.5 ($P>0.5$). The $P$ value was calculated with the chi-square test. (**b**) Representative $Xlf^{+/-}Paxx^{+/-}$ and littermate $Xlf^{-/-}Paxx^{-/-}$ embryos at E14.5. The length of the scale bar is 1 cm. (**c**) Haematoxylin and eosin (H&E) staining of E14.5 embryonic brains from littermate $Xlf^{-/-}$ or $Xlf^{-/-}Paxx^{-/-}$ mice. The ventricular zone (VZ) contains the proliferating cells. The intermediate zone (IZ) and cortical plate (CP) contain post-mitotic neurons. Black arrowheads denote condensed nuclei. The scale bar in **c** stands for 100 μm. (**d**) Immunostaining of E14.5 embryonic brain using antibodies against Cleaved (activated) Caspase3. Brown staining represents apoptotic cells. The scale bar in **d** stands for 100 μm. (**e**) The frequency of condensed nuclei per 1/20 of × 400 field in E14.5 brain were quantified. The data represent nine different fields from at least two biological repeats of each genotype. The $P$ value was calculated with the chi-square test.

only $2.0\pm0.1\%$ in $Xlf^{+/+}Paxx^{+/+}$ (WT), $3.4\pm0.3\%$ in $Paxx^{-/-}$ and $9.6\pm0.1\%$ in $Xlf^{-/-}$ primary MEFs (Fig. 4c and Supplementary Fig. 3D). T-FISH can distinguish two types of cytogenetic abnormalities[31]: (1) chromosomal breaks involving both sister chromatids, which reflect repair defects occurring in the G1 phase of the cell cycle, and (2) chromatid breaks involving only one chromatid, which reflect damage that occurs after DNA replication. Of note, almost all of the cytogenetic abnormalities in $Xlf^{-/-}Paxx^{-/-}$ cells were chromosomal breaks (Fig. 4c and Supplementary Fig. 3D), similar to the cytogenetic defects observed in cells deficient for other core NHEJ factors. Together, these findings reveal a critical complementary function between PAXX and XLF in maintaining genomic stability during normal cell growth and in response to IR or etoposide challenges. The skew towards

chromosomal breaks together with the presense of IR and etoposide hypersensitivity strongly suggest defects in the NHEJ pathway in $Xlf^{-/-}Paxx^{-/-}$ cells.

**End-joining is abrogated in $Xlf^{-/-}Paxx^{-/-}$ cells.** To further characterize the function of PAXX and XLF in NHEJ, we used CRISPR/Cas9 technology to delete the entire murine *Paxx* gene from WT or $Xlf^{-/-}E\mu$-$Bcl2^{+}$ Abelson Murine Leukemia Virus (A-MuLV)-transformed pre-B cell lines (v-abl cells) (Fig. 5a). These CRISPR/Cas9-deleted cells are referred to as $Paxx^{\Delta/\Delta}$, to distinguish them from germline $Paxx^{-/-}$ mice and cells. Successful deletion of the *Paxx* gene was monitored by Southern blotting (Fig. 5b) and the absence of PAXX protein was confirmed by western blot analyses (Fig. 5c).

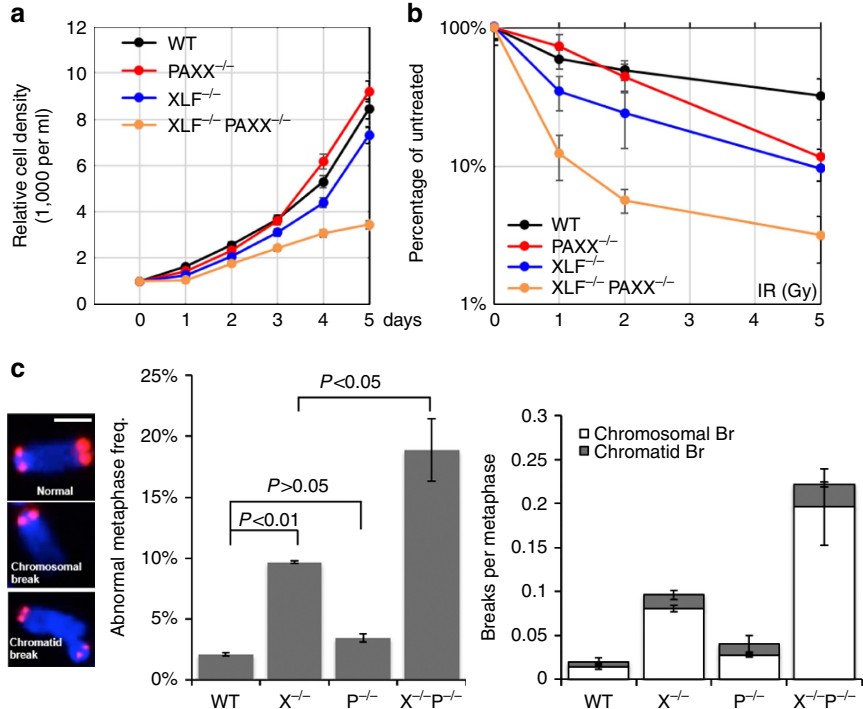

**Figure 4 | Proliferation defects and IR hypersensitivity of $Xlf^{-/-} Paxx^{-/-}$ MEFs.** (**a,b**) Proliferation and IR sensitivity assays were performed using P2 primary MEFs with different genotypes. Three independent repeats were performed. The average and s.d. were plotted. The Y axis is the relative cell number measured by hemocytometer (for **a**) or the fluorescence-based nucleotide dye (CyQuant Cell Proliferation Assay, Invitrogen, CA) (for **b**). (**c**) Left: the examples of normal chromosome, chromosomal breaks and chromatid breaks (one of two sister chromatids is broken). Middle: frequency of metaphases with one or more cytogenetic abnormalities in P1 primary MEFs of different genotypes. At least two independently derived lines were assayed for each genotype. The scale bar stands for 1 μm. Right: the frequency of chromatid (grey box) and chromosomal (white box) breaks measured by T-FISH analyses in P1 primary MEF of different genotypes. The average and s.d. of at least three independent biological repeats were shown. $X^{-/-} = Xlf^{-/-}$, $P^{-/-} = Paxx^{-/-}$ and $X^{-/-}P^{-/-} = Xlf^{-/-} Paxx^{-/-}$. The P value was calculated by two-tailed Student's t-test. The raw data were summarized in Supplementary Fig. 3D.

Consistent with the results from primary MEFs (Fig. 4b), v-abl kinase-transformed $Xlf^{-/-} Paxx^{\Delta/\Delta}$ pre-B cells were hypersensitive to IR, similar to the $Xrcc4^{-/-}$ control, while $Paxx^{\Delta/\Delta}$ pre-B cells are, at most, moderately sensitive to IR (Fig. 5d). Also consistent with the data from MEFs, $Xlf^{-/-} Paxx^{\Delta/\Delta}$ pre-B cells display higher levels of spontaneous genomic instability, especially chromosomal breaks (Fig. 5e and Supplementary Fig. 3E).

To measure NHEJ function specifically, we utilized the chromosomal V(D)J recombination assay[15]. In this system, v-abl kinase inhibitor STI571 (also called Gleevec) induces a G1 phase cell cycle arrest in the v-abl cells that allows accumulation of RAG protein, which initiates efficient V(D)J recombination at a chromosomal integrated pMX-INV inversional substrate[15]. pMX-INV contains an inverted green fluorescent protein (GFP) cassette flanked by recombination signal sequences (RSS, triangles in Fig. 6a) that can be cleaved and inverted back to the same orientation as the promoter by RAG-mediated recombination reaction. NHEJ-mediated repair and the formation of both SJs and CJs are necessary and sufficient for the expression of GFP protein (Fig. 6a)[3,15]. Using this assay, we found that the integrated pMX-INV substrate was rearranged robustly in two isogenic clones of $Paxx^{\Delta/\Delta}$ cells derived from parental WTInv4 cells (pMX-INV at the same genomic locus in both WT and $Paxx^{\Delta/\Delta}$ cells), as measured by the appearance of CJs in Southern blots (Fig. 6b and Supplementary Fig. 4B) and the accumulation of GFP expressing cells by FACS (Supplementary Fig. 4A). Thus, PAXX itself is not required

for chromosomal V(D)J recombination. In contrast, while parental $Xlf^{-/-}$ cells can be induced to express GFP efficiently, $Xlf^{-/-} Paxx^{\Delta/\Delta}$ cells failed to express GFP upon STI571 induction and accumulated both 3'-CEs and the small CEs–SEs fragment similar to those seen in $Xrcc4^{-/-}$ cells, indicative of end-ligation defects (Supplementary Fig. 4A and Fig. 6c). Ectopic expression of either PAXX or XLF protein in $Xlf^{-/-} Paxx^{\Delta/\Delta}$ B cells restored V(D)J recombination, supporting the specificity of $Paxx$ deletion by CRISPR/Cas9 (Fig. 6d, Supplementary Fig. 4C). Moreover, $Xlf^{-/-} Paxx^{\Delta/\Delta}$ B cells, similar to $Xrcc4^{-/-}$ cells, also failed to repair chromosomal DSBs generated by Ppo1 endonuclease, indicating that their end-ligation defects are not limited to V(D)J recombination (Fig. 6e). These findings, together with the severe neuronal apoptosis of $Xlf^{-/-} Paxx^{-/-}$ mice, reveal a critical function for PAXX in NHEJ-mediated end-ligation that is otherwise masked by XLF.

**PAXX and XLF have distinct functions during NHEJ.** Given their structural similarities, it is conceivable that PAXX and XLF are functionally equivalent. However, ATM (KU55933) or DNA-PK (NU7441) kinase inhibitors did not block the formation of CJ products or the expression of GFP in $Paxx^{-/-}$ cells, despite the fact that $Xlf^{-/-}$ pre-B cells require both ATM and DNA-PK kinase activity for efficient V(D)J recombination (Figs 6b and 7a, and Supplementary Fig. 4A)[12,28]. Moreover, overexpression of PAXX was not sufficient to rescue the

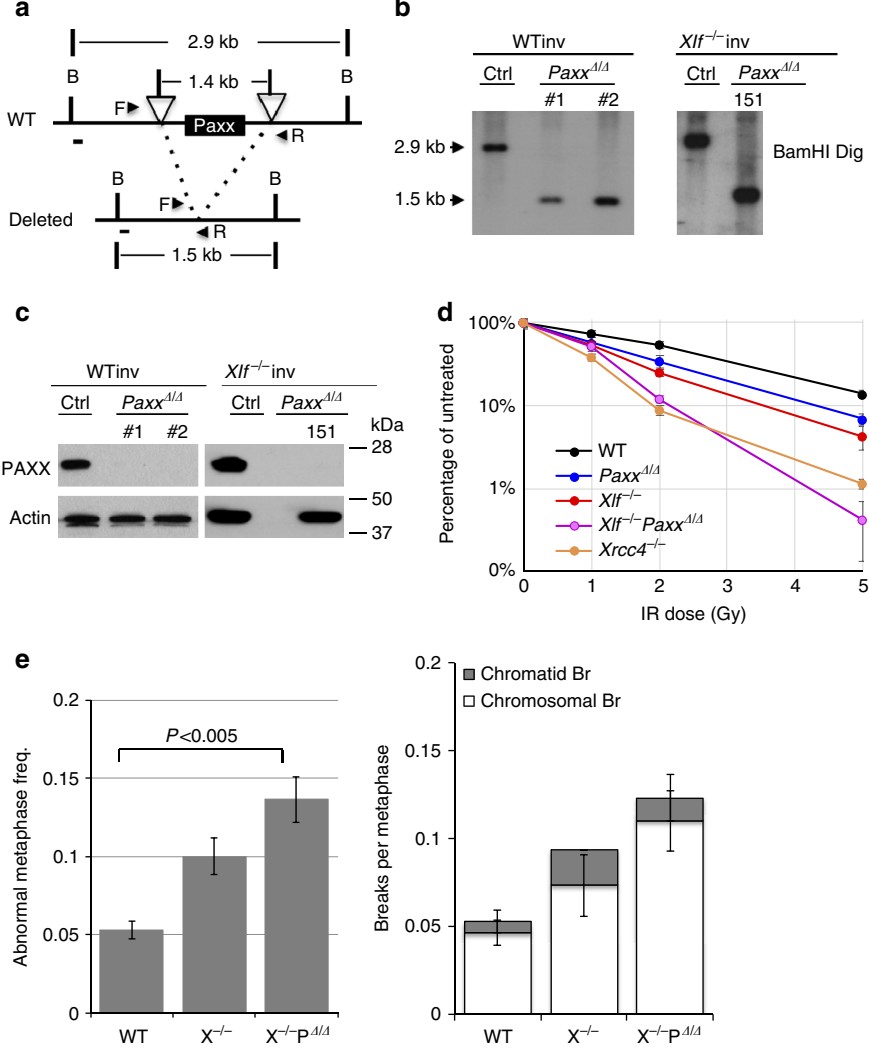

**Figure 5 | Generation and characterization of isogenic $Xlf^{-/-} Paxx^{\Delta/\Delta}$ pre-B cell lines.** (**a**) Scheme for murine *Paxx* locus and CRISPR/Cas9 mediated knockout strategy. The open triangles indicate the location of the guided RNA. The horizontal arrows indicate the location of the primers used for primary screen. Southern blot confirmation was performed using BamHI-digested DNA with the mPaxx probe (show as thick black line). (**b,c**) Southern blot and western blot assays confirmed the homozygous deletion of *Paxx* and the absence of Paxx protein in WT or $Xlf^{-/-}$ v-abl transformed pre-B cells with inverted substrates. (**d**) IR sensitivity of WT, $Paxx^{-/-}$, $Xlf^{-/-}$, $Xlf^{-/-} Paxx^{\Delta/\Delta}$ and $Xrcc4^{-/-}$ pre-B cells measured, determined by cell counting. At least triplicates were plated and counted for each genotype at each dosage. (**e**) Left: frequency of metaphases with cytogenetic abnormalities of WT, $Xlf^{-/-}$ or $Xlf^{-/-} Paxx^{\Delta/\Delta}$ v-abl transformed pre-B cells. At least three independent experiments were performed. The data represent the average and the s.d. of the three repeats. Right: the frequency of chromatid (grey box) and chromosomal (white box) breaks measured by T-FISH analyses in the pre-B cells. The raw data were summarized in Supplementary Fig. 3E. $X^{-/-} = Xlf^{-/-}$, $P^{\Delta/\Delta} = Paxx^{\Delta/\Delta}$ and $X^{-/-} P^{\Delta/\Delta} = Xlf^{-/-} Paxx^{\Delta/\Delta}$.

end-ligation defects caused by ATM inhibition in Xlf-deficient cells (Supplementary Fig. 4A). Together these findings suggest that PAXX cannot replace XLF, and XLF and PAXX likely have distinct functions during NHEJ.

Given $Xlf^{-/-}$ cells require ATM as well as PAXX for end-ligation, we asked whether PAXX affects ATM activation. IR- or camptothecin (CPT)-induced phosphorylation of specific ATM substrate KAP1 was unaffected in $Paxx^{-/-}$ cells (Fig. 7b and Supplementary Fig. 5B). Moreover, while ATM is required for chromosomal end-ligation and dispensable for end-ligation of extra-chromosomal (episomal) DNA in XLF-deficient cells[12], both chromosomal and extra-chromosomal SJs formation decreased nearly 100-fold in $Xlf^{-/-} Paxx^{\Delta/\Delta}$ cells, in comparison with WT or $Xlf^{-/-}$ or $Paxx^{\Delta/\Delta}$ single deficient cells (Fig. 7c and Supplementary Fig. 5A). Together, these results suggest a direct function of

PAXX in DNA end-ligation that is distinct from both ATM and XLF.

**PAXX promotes the accumulation of KU at DNA ends *in vivo*.** KU, especially the conserved N-terminal regions required for hetero-dimerization, is conserved through evolution and is essential for DNA end-ligation *in vivo* and *in vitro*. PAXX binds to KU directly[16,17] and is rapidly recruited to DNA damage sites. So we tested whether KU is essential for the recruitment of PAXX to the DSB *in vivo*. GFP-PAXX was efficiently recruited to the site of laser-induced DNA lesions in $Ku80^{+/+}$, but not in $Ku80^{-/-}$ MEFs (Fig. 7d). Moreover both full-length KU80 and a truncated KU80 lacking the flexible C-terminal domain ($\Delta$CTD) implicated in DNA-PKcs binding, fully restored PAXX recruitment (Fig. 7d and Supplementary Fig. 5C), suggesting PAXX binds to the conserved N-terminal region

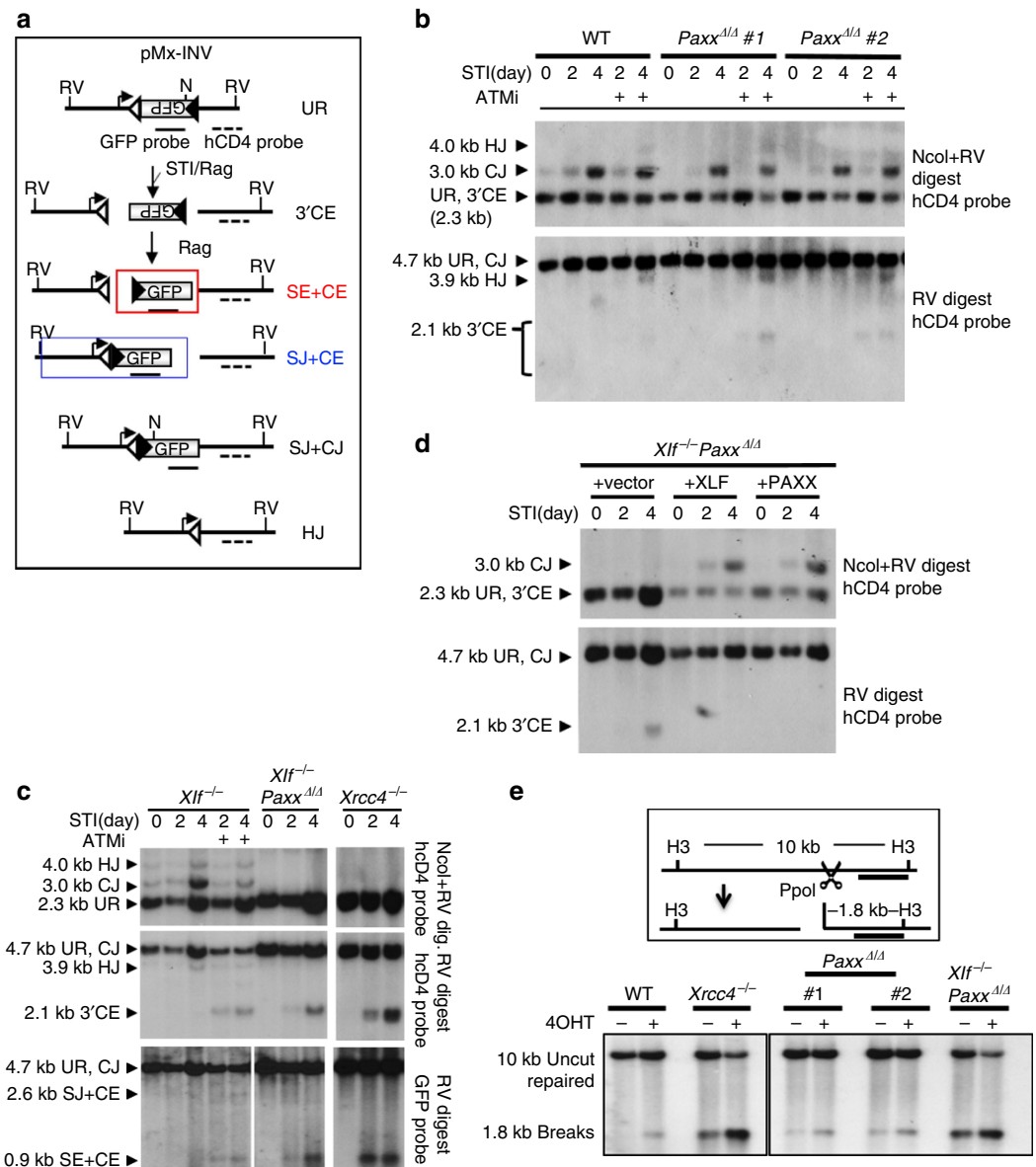

**Figure 6 | V(D)J recombination and end-joining defects in *Xlf*[−/−]*Paxx*[Δ/Δ] cells.** (**a**) Schematic of pMX-INV V(D)J recombination substrates[15] (diagram modified from Zha *et al.*[12]). The unrearranged substrate (UR), hybrid joins (HJs), coding/signal end (CE/SE) intermediates and coding/signal joins (CJs/SJs) are diagrammed. The recombination signal sequence (RSS, triangle), GFP probe (solid lines) and hCD4 probe (dash lines) are indicated. Positions of EcoRV (RV) sites and NcoI (N) sites are shown. The red box shows the isolated SE–CE fragments resulting from complete end-ligation defects. The blue box shows the SJ–CE fragments resulting from isolated CJ formation defects (for example, hairpin opening defects in *DNA-PKcs*[−/−] cells)[3]. (**b–d**) Southern blot analyses of pMX-INV rearrangement products and intermediates. The digestion and probes are marked on the side of the blots. The WT and the two independently derived *Paxx*[Δ/Δ] lines in **b** have the pMX-INV substrates integrated at the same genomic location. And the *Xlf*[−/−] and isogenic *Xlf*[−/−]*Paxx*[Δ/Δ] derivative have their pMX-INV integrated in the same genomic locus. *Xrcc4*[−/−] cells were included as a control. (**e**) Diagram of PpoI-mediated cleavage in one mouse genomic DNA site on chromosome 1. H3 = Hind III. The dark line indicates the position of the probe designed to detect DNA ends. Lower: Southern blot analyses for unrepaired ends from representative cells with HindIII-digested DNA.

of KU. Given the importance of KU in end-ligation, we next asked whether PAXX affects KU dynamics *in vivo* and found that loss of PAXX, but not the loss of XLF, significantly reduced the intensity of GFP-KU70 at DSBs *in vivo* (Fig. 7e). Correspondingly, the intensity of GFP-KU70 also reduced significantly in *Xlf*[−/−]*Paxx*[−/−] MEFs, although consistently higher than in *Paxx*[−/−] cells, likely caused by the continuous recruitment of KU due to persistent breaks in *Xlf*[−/−]*Paxx*[−/−] cells (Fig. 7e). Meanwhile, the level of GFP-Lig4 is markedly reduced in *Xlf*[−/−] MEFs and nearly absent in *Xlf*[−/−]*Paxx*[−/−] cells (Fig. 7f). At the same time, the recruitment of GFP-Lig4 is

at most moderately affected in *Paxx*[−/−] cells, which is likely secondary to the KU-recruitment defects (Fig. 7f). Together these data suggest a model in which PAXX promotes robust recruitment of KU, while XLF enhances the recruitment of Lig4, and together they ensure efficient end-ligation through distinct yet complementary mechanisms (Fig. 7g).

## Discussion

PAXX was recently proposed to function as a NHEJ factor on the basis of its structural homology to XRCC4 and XLF, and the IR sensitivity of PAXX-depleted human tumour lines.

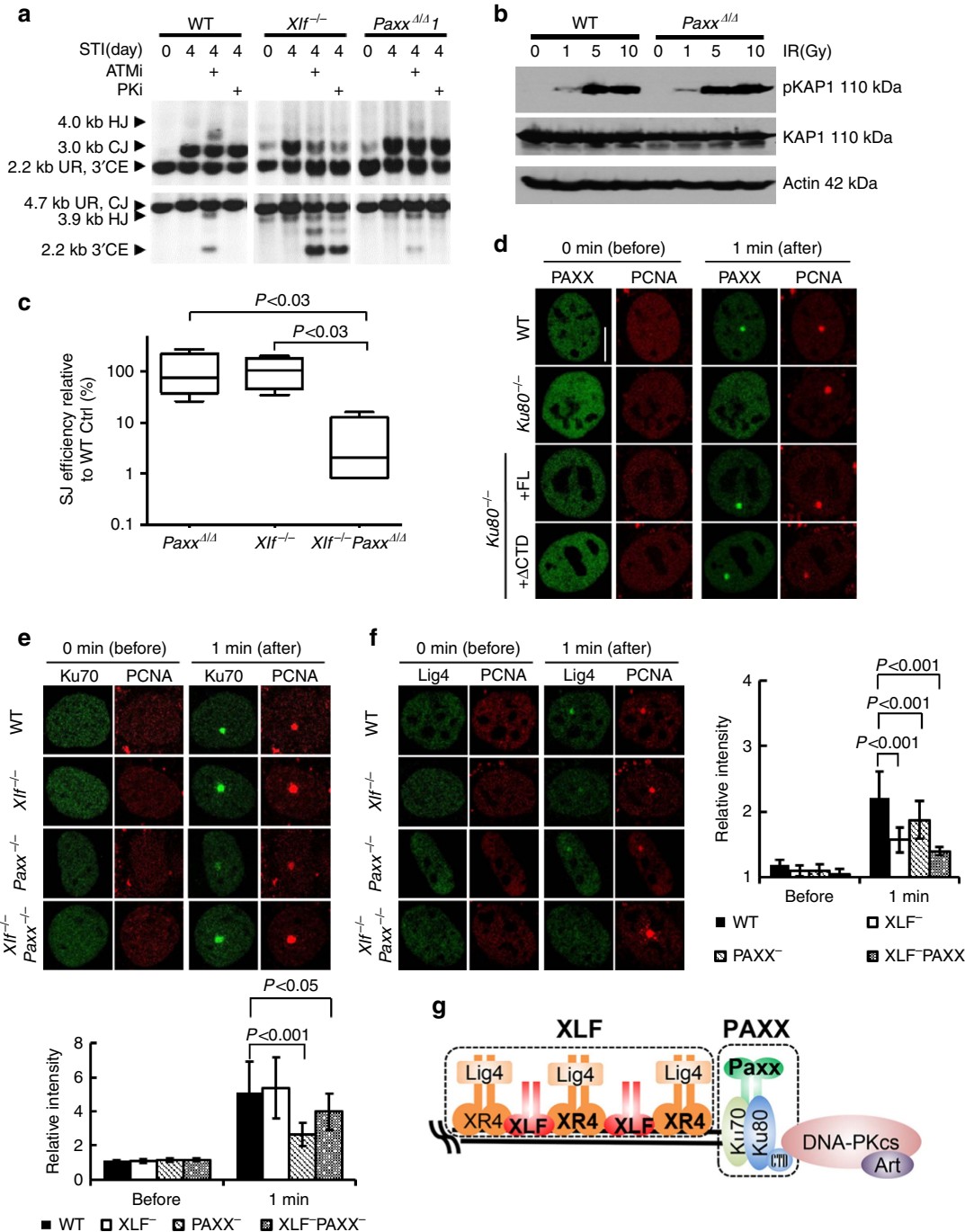

**Figure 7 | PAXX and XLF have distinct functions during NHEJ.** (**a**) Southern blot analyses of the product and intermediates of pMX-INV V(D)J recombination reporter in WT, *Xlf⁻/⁻* and *Paxxᐞ/ᐞ* cells with or without ATM kinase inhibitor (KU55933, 15 μM) or DNA-PKcs inhibitor (NU7441, 5 μM) added 1 h before STI571 (3 μM). (**b**) Western blot for phosphorylated KAP1, total KAP1 and β-actin in total lysate from pre-B cells immediately after irradiation (at 1, 5 or 10 Gy). (**c**) Extrachromosomal SJ formation efficiencies in WT, *Paxxᐞ/ᐞ*, *Xlf⁻/⁻* and *Xlf⁻/⁻Paxxᐞ/ᐞ* Pre-B cells. Each box represents the average and s.d. of four independent repeats. Box centre line is plotted at the median and the box extends from the 25th to 75th percentiles. Whiskers go down to the smallest value and up to the largest value. The raw data were summarized in Supplementary Fig. 5A. The *P* value was calculated by two-tailed Student's *t*-test. (**d**) Laser-induced recruitment of GFP-PAXX or DsRed-PCNA in immortalized *Ku80⁻/⁻* MEFs with or without ectopic expression of full-length (FL) Ku80 or C-terminal truncated Ku80 (ΔCTD). Ten cells were tested for each genotype and representive images are shown here. (**e**) Laser-induced recruitment of GFP-Ku70 in immortalized WT, *Xlf⁻/⁻*, *Paxx⁻/⁻* and *Xlf⁻/⁻ Paxx⁻/⁻* MEFs 1 min after damage. *P*<0.001 between *Paxx⁻/⁻* and WT cells, *P*<0.05 between *Xlf⁻/⁻ Paxx⁻/⁻* and WT cells, two-way ANOVA (**f**) Laser-induced recruitment of GFP-Lig4 in immortalized WT, *Xlf⁻/⁻*, *Paxx⁻/⁻* and *Xlf⁻/⁻ Paxx⁻/⁻* MEFs 1 min after damage. *P*<0.001 between *Xlf⁻/⁻* and WT cells, *P*<0.01 between *Paxx⁻/⁻* and WT cells, *P*<0.001 between *Xlf⁻/⁻ Paxx⁻/⁻* and WT cells, two-way ANOVA. The bar graphs of **e**,**f** represent the average and error bars stand for s.d. (see Methods for details), same experiments were repeated for at least two times. The scale bar in **d** stands for 10 μm. (**g**) Working model. In this model, we propose that XLF (diagrammed as red bean sprout shape) enhances the efficient accumulation of Lig4 in complex with XRCC4 at the DNA ends, while PAXX (diagrammed as green bean sprout shape) promotes the accumulation of KU at the DNA ends. Together PAXX and XLF promote efficient DNA end-ligation through distinct mechanisms. XR4, XRCC4; Art, Artemis.

Unexpectedly, we found that PAXX is largely dispensable for NHEJ in mice under physiological conditions. Lymphocyte development, including both V(D)J recombination and CSR, as well as the general NHEJ repair activity required for maintenance and expansion of peripheral lymphocytes, are not affected by the loss of PAXX. Accordingly, PAXX-deficient primary MEFs proliferated well in culture and did not accumulate significantly more spontaneous chromosomal breaks than WT controls. Yet PAXX-deficient primary MEFs or v-abl kinase-transformed B cells displayed a moderate, but significant, sensitivity to IR. These data indicate that PAXX serves as a genomic caretaker that is largely dispensable under physiological conditions, and instead functions to ensure genomic integrity under stresses or in selective genetic backgrounds. One of these conditions would be XLF deficiency. While neither PAXX nor XLF alone is essential for murine development or lymphocyte V(D)J recombination, $Paxx^{-/-}$ mice require XLF and $Xlf^{-/-}$ mice require PAXX for end-ligation. As such, $Xlf^{-/-}Paxx^{-/-}$ mice died during late embryonic development with severe neuronal apoptosis similar to $Lig4^{-/-}$ or $Xrcc4^{-/-}$ mice[10,11]. $Xlf^{-/-}Paxx^{-/-}$ primary MEFs accumulate high levels of chromosomal breaks, leading to cell cycle arrest, and eventually fail to thrive. $Xlf^{-/-}Paxx^{\Delta/\Delta}$ lymphocytes cannot complete V(D)J recombination and thus accumulate unrepaired SEs and CEs. Notably, both chromosomal and extra-chromosomal end-joining were abolished in $Xlf^{-/-}Paxx^{\Delta/\Delta}$ cells, supporting a role of PAXX in DNA end-ligation directly. Accordingly, we found that PAXX, but not XLF promotes the maximal recruitment of KU to DNA ends in vivo. Taken together, our findings identified distinct, yet complementary functions of two core NHEJ factors—PAXX and XLF in end-ligation and revealed an intricate regulation of the NHEJ pathway. Consistent with our findings, two recent studies using B-cell lines with CRISPR-Cas9 knockout of PAXX and XLF also revealed their redundant function in end-joining[32,33].

PAXX shares structural similarity with both XRCC4 and XLF. Unlike XRCC4 (refs 34,35), both PAXX and XLF have relatively short coiled-coil stalks that cannot bind to Lig4 directly and are not absolutely required for end-ligation. In vivo, loss of either XLF or PAXX has negligible effects on the survival of post-mitotic neurons (Fig. 3e), proliferation of primary MEFs (Fig. 4a), or chromosomal V(D)J recombination in developing lymphocytes (Fig. 6). While these phenotypes are strikingly similar, there are notable differences. Loss of XLF reduced the cellularity of peripheral lymphoid organs and caused a ~50% reduction in CSR[30], while PAXX deficiency has no measurable impact on CSR or the cellularity and growth of lymphoid organs (Fig. 2)[32]. In addition, XLF-deficient MEFs exhibit spontaneous genomic instability, while PAXX-deficiency did not lead to statistically significant increases in genomic instability (Fig. 4). Moreover, PAXX-deficient MEFs exhibit little, if any, sensitivity to IR, in contrast to the moderate IR sensitivity of XLF-deficient cells (Figs 4 and 5). Most strikingly, ATM and DNA-PKcs kinase activities are required for chromosomal end-joining in XLF-deficient cells, but are dispensable in PAXX-deficient cells (Fig. 7). At the molecular level, XLF forms filamental structures with XRCC4, which promotes Lig4 re-adenylation in vitro[36]. Here we found that XLF also promotes efficient recruitment of Lig4 in vivo without affecting KU (Fig. 7f). In contrast, PAXX binds directly to DNA-bound KU and promotes accumulation of KU at DSBs in a reciprocal manner (Fig. 7e). Based on these data, we proposed that while XLF and PAXX have important complementary activities, they function in distinct protein complexes, and through different mechanisms to ensure end-ligation (Fig. 7g).

So what is the role of PAXX in end-joining? Like DNA-PKcs, PAXX interacts with DNA bound KU, but not free KU. But in contrast to DNA-PKcs, which interacts with the C-terminal tails of Ku80, PAXX requires the core of KU, but not the Ku80 C-terminal domain, for its recruitment to DNA damage sites (Fig. 7). We found that in the absence of PAXX, the maximum intensity of GFP-Ku70 at the DNA damage site reduced significantly (Fig. 7). Given KU is essential for the initial recruitment of PAXX, we postulate that PAXX might stabilize KU at the DNA ends, rather than affecting the initial recruitment of KU to DNA. Since $Ku$ deletion rescues the embryonic lethality of Lig4-deficient mice[37], presumably by allowing the alternative end-joining (A-EJ) pathway to access the DNA ends, this role of PAXX in stabilizing KU might explain why knockout of $Paxx$ in $Xrcc4^{-/-}$ chicken DT40 cells partially rescues the severe IR sensitivity[17]. If so, why do $Xlf^{-/-}Paxx^{-/-}$ embryos die of severe neuronal apoptosis? We note that the intensity of Lig4 foci is lowest in $Xlf^{-/-}Paxx^{-/-}$ cells, while the intensity of Ku foci is actually higher in $Xlf^{-/-}Paxx^{-/-}$ cells than in $Paxx^{-/-}$ cells (Fig. 7e,f), likely due to the continuous recruitment of KU in the absence of DNA repair. This recruitment pattern in $Xlf^{-/-}Paxx^{-/-}$ cells (the presence of KU without Lig4) is similar to that found in the $Lig4^{-/-}$ cells, where both NHEJ (due to Lig4 deficiency) and A-EJ (due to the presence of KU) are limited, and might explain the severe end-ligation defects and embryonic lethality of $Xlf^{-/-}Paxx^{-/-}$ mice. In this context, the sensitivity to genotoxic agents or spontaneous damages would be determined by a balance and availability of NHEJ and A-EJ to given types of breaks. Given the robust NHEJ in $Paxx^{-/-}$ mice, PAXX-dependent stabilization of KU at DNA ends is clearly not essential for NHEJ in otherwise WT cells. This function of PAXX becomes critical when XLF is also absent in the cells. XLF-deficient cells also require ATM, DNA-PKcs, 53BP1 and H2AX for end ligation[5,12–14,24,28]. In addition, the C terminus of the RAG2 protein is required for chromosomal V(D)J recombination specifically in XLF-deficient cells[28]. ATM and its substrates are thought to increase DNA end stability while the C terminus of RAG has been proposed to facilitate end-ligation in XLF-deficient cells during V(D)J recombination by stabilizing the post-cleavage complex embracing the DNA ends[15,38]. Therefore, it is conceivable that PAXX might also facilitate end-ligation by promoting end stability and synapsis similar to RAG2 during V(D)J recombination and to ATM/DNA-PK through their chromatin-bound substrates. Consistent with their similar function, end-ligation in the PAXX-deficient cells is not further affected by loss of ATM or DNA-PK kinase activity or in RAG-independent breaks (Figs 6e and 7a)[32,33]. KU has been proposed to have synaptic function through the recruitment and intermolecular phosphorylation of DNA-PKcs[39,40]. Given the direct interaction between PAXX and KU and the role of PAXX in the accumulation of KU at DSBs in vivo (Fig. 7d,e), it is tempting to speculate that PAXX modulates the stability of the DNA-PKcs/KU-DNA complex to promote end-synapsis and support end-ligation in XLF-deficient cells. Consistent with this model, inhibiting DNA-PK kinase activity has no additional impact on end-ligation in $Paxx^{-/-}$ cells (Fig. 7a).

PAXX is the first bona fide NHEJ factor that was not firstly identified in patients or as naturally occurring mutations in another organism. Patients with loss-of-function mutations of Lig4, XRCC4, XLF, Artemis and DNA-PKcs develop SCID and variable levels of microcephaly[1,2]. Given the severe IR sensitivity observed in PAXX-depleted human tumour lines, patients with PAXX deficiencies have been expected. Yet, the mild phenotype of $Paxx^{-/-}$ mice suggests that PAXX deficiency might not be readily recognized. While the precise reason for the

strong dependence on PAXX in NHEJ in human cancer cell lines and the dispensable role of PAXX in mice *in vivo* is yet to be determined, species differences are unlikely to be the only explanation. While in U2OS cells and 293H cells, PAXX deficiency causes severe IR sensitivity, loss of PAXX causes only moderate, if any, IR sensitivity in human colorectal cancer HCT116 cells and 293 cells[17,24]. It is possible that genetic heterogeneity, including the availability of other repair factors might contribute to this variability. So what would be the physiologic function of the evolutionarily conserved PAXX gene? One possibility is that PAXX may promote tumour suppression or prevent premature aging in long-living organisms. While lymphocyte numbers are only moderately reduced in young XLF-deficient mice, old XLF-deficient mice (24 mos.) display rapid reduction of lymphocyte numbers owing to decreased renewal and function of the hematopoietic cells[41]. If a similar phenotype is observed in older $Paxx^{-/-}$ mice, then PAXX-deficient patients, if they exist, might present premature ageing phenotypes such as late-onset immunodeficiency or myeloid proliferative diseases. Except rare truncating mutations in the C-terminal KU-binding region of PAXX, the *PAXX (C9Orf142)* gene is amplified and overexpressed in several human cancers (cBioPortal), suggesting that PAXX function might modulate the therapeutic responses to genotoxic cancer therapies, such as radiation.

## Methods

**Mice.** To generate $Paxx^{-/-}$ mice, a targeting construct was made to replace part of Exon 1 (after ATG) and exons 2–7 of the Paxx gene with a neomycin resistance gene. To generate the Paxx targeting construct, 5′ (∼2 kb) and 3′ (3.4 kb, with HindIII linker) arms were PCR amplified from 129/Sv murine ES cell DNA with high fidelity DNA polymerase (Phusion, NEB), cloned into pGEMT shuttle vectors (Promega) and sequenced (Genewiz). The primers used for the PCR are listed in Supplementary Table 2. The 5′arm and 3′arm were then released and subcloned to the pEMC targeting vector with NeoR flanked by frt sites using HindIII or NotI respectively. The final vector was validated by sequencing and restriction digestions. ClaI-linearized pEMC-PAXX plasmid was electroporated into CSL3 murine ES cells (129/Sv) and selected with G418 (300 μg ml$^{-1}$) and Ganciclovir (Sigma G2566, 2 μM) for 7 days. The correct targeted clones were identified by Southern blotting analyses using EcoRV-digested genomic DNA and a 3′ genomic probe as outlined in Fig. 1a (the primers used to generate the probe are listed in Supplementary Table.2). The WT band is ∼4.7 kb, and the targeting destroys one of the EcoRV site in Exon 3, resulting in a ∼7.0 kb band. A total of eight independent targeted ES cells were obtained and two were injected for germline transmission and yielded identical phenotypes. Genotyping was performed with primers listed in Supplementary Table 2. Primers used to generate the mPAXX probes are also listed in Supplementary Table 2. $Xlf^{-/-}$ mice[7,42] were bred to $Paxx^{-/-}$ mice to generate $Xlf^{-/-}Paxx^{-/-}$ mice and $53BP1^{-/-}$ mice[43] (Supplementary Fig. 1D) were used as a control for thymocyte development defects. Both male and female mice were used in all experiments. All animal work has been conducted in a specific pathogen-free facility and all the procedures were approved by Institutional Animal Care and Use Committee (IACUC) at Columbia University Medical Center.

**Lymphocyte development and recombination.** Single-cell suspensions were prepared from lymphocytes of thymus (Thy), bone marrow (BM), spleen (Spl) and lymph node (LN) from 7-week-old mice of the described genotypes and ∼1 × 10$^5$ cells were stained using fluorescence-conjugated antibodies as indicated before analysed by flow cytometry. For Class Switch Recombination assay, single-cell suspensions of spleen cells were sorted with CD43 magnetic beads (MACS, Miltenyi), and B cells were cultured at a density of 5 × 10$^5$ cells per ml in RPMI medium supplemented with 10% FBS and 25 ng ml$^{-1}$ of LPS plus 25 ng ml$^{-1}$ of IL-4 (R&D). Cultured cells were maintained daily at a density of 1 × 10$^6$ cells per ml. Cells were collected on various days for flow cytometry. Single-cell suspensions from spleens were prepared according to the standard methods from mice 6 to 12 weeks old. Cells from cultures on day 4 were washed twice in PBS plus 2% FBS and were stained with various antibodies conjugated with fluorescein isothiocyanate (IgG1, BD Pharmingen and B220, eBioscience). Flow cytometry was performed on a FACS Calibur flow cytometer (BD Bioscience) and data were processed using FlowJo software package.

**Endogenous Vbeta14DJbeta1 junction analyses.** Vbeta14DJbeta1 coding joins were amplified from total thymocyte DNA with the following primers: Vbeta14F:

5′-AGAGTCGGTGGTGCAACTGAACCT-3′ and Jb1.2 R: 5′-CCTGACTT CCACCCGAGGTTC-3′. Two amplicons were identified from each sample corresponding to rearrangement involving Jβ1.1 or Jβ1.2. The two amplicons were gel-purified separately and cloned into a pGEM-T Easy Vector (Promega) following the manufacturer's protocol. Individual clones were sequenced with the built-in T7 primer site and aligned with the Vβ14 and Jβ sequence (accession no. AE000665). Sequence analyses identified the V, D, J coding region first, then identified Palindromic (P) element and non-template nucleotide addition (N). The number of nucleotide deletions for the V–D junction and D–J junctions were also calculated (Supplementary Table.1).

**Histological analyses.** Embryo heads were fixed in 4% paraformaldehyde, paraffin embedded, serially sectioned (sagittal, 5 μm) and stained with hematoxylin and eosin (H&E) or cleaved Caspase-3 (Cat#9664, Cell Signaling) in the standard histopathology core of Columbia University.

**Proliferation and chemosensitivity.** Primary MEFs isolated from E14.5 embryos were designated passage 0 (P0) and were cultured in DMEM medium (GIBCO, Life Technology) containing 15% fetal bovine serum, 100 mM L-glutamine, and penicillin–streptomycin on a gelatinized plate until they reached confluence. Equal numbers of cells (1 × 10$^4$) were plated into a well of a six-well plate. The medium was changed daily and MEFs were counted at different time points in triplicate using a hemocytometer.

For IR, etoposide and HU sensitivity assays, cells were plated on 96-well plates 24 h before irradiation or the addition of genotoxic agents (etoposide: 0.01, 0.1 and 1 μM; hydroxyurea: 20, 200 and 2,000 mM. The drug was washed away 24 h after the treatment). The relative cell density was determined 7 days after irradiation or drug treatment by fluorescence nucleotide dye CyQuant (MEFs) or by hemocytometer (v-abl transformed pre-B cells).

For cell cycle analyses, proliferating primary MEF cells were incorporated with 10 μM BrdU for 30 min and fixed and then stained with Anti-BrdU FITC (Cat# 11-5071-41, eBioscience) and propidium iodide (PI) and analysed by flow cytometry

**Metaphase preparation and telomere-FISH staining.** The metaphase was prepared as previously described[3]. Briefly, cells were cultured for 6 (primary MEF) or 3 (v-abl transformed pre-B cells) hours with colcemid (KaryoMAX Colcemid Solution, GIBCO) to the final concentration of 100 ng ml$^{-1}$. The treated cells were then collected and swollen using hypotonic solution at room temperature and fixed (fresh fixative made by 3:1 v/v methanol: acetic acid). Metaphase spreads were obtained by dropping fixed cells onto pre-cleaned slides. Slides were fixed in 4% (w/v) formaldehyde/PBS, followed by three washes with PBS and digestion with pepsin/PBS (0.1%; Sigma). The slides were then washed three times in PBS and dehydrated through ethanol, serially. A Cy3-labelled (CCCTAA)$_3$ peptide nucleic acid (PNA) probe (customer synthesized, Biosynthesis Inc.) was used to hybridize the metaphases under denaturing conditions (heating for 3 min at 80 °C on a heat block) and incubated in a dark humidity chamber for 2 h. The slides were washed and dehydrated in ethanol and air dried. DNA was counterstained with DAPI. A minimum of 100 cells with telomere signals were captured by Metafer4 using Metasystems automatic Metaphase search and scan system equipped with a Plan Fluor Nikon Lens (× 63/1.30 Oil, Japan) and counted for breaks.

**Cas9-mediated deletion of *Paxx*.** A pair of guide RNAs (gRNA) targeting the mouse *Paxx* gene were designed and cloned into the pX330 CRISPR plasmid (generously provided by Dr Feng Zhang through Addgene) (gRNA1: 5′-CACC GCTAAGGTGTTCGCTCGGCGG-3′; gRNA2: 5′-CACCGCAGTTTATTTGACG GAGAA-3′). The pair of gRNA containing plasmids were electroporated into v-abl kinase-transformed WT or $Xlf^{-/-}$ v-abl transformed pre-B cells using the 4D Nucleofector apparatus (Buffer SF, program DN100, Lonza, Walkersville, MD). The cells were plated for single clones, and PCR (5′-ATTGAAGAGCGGCAGA TATGT-3′ and 5′-ACGCAGAATCAACACAGTAGGT-3′) was performed on each single clone to identify the deletion. When necessary, heterozygous targeted clones were targeted a second time to remove the second allele. Complete deletion was verified by PCR, Southern blot and western blot.

**Construction of plasmids.** pBMN-IRES-GFP was purchased from Addgene and the GFP was replaced by a truncated humanCD2 fragment to generate pBMN-IRES-hCD2. A Flag sequence was then inserted in front of the multiple cloning site of pBMN-IRES-hCD2 to generate pBMNFlag-IRES-hCD2. Full-length mouse PAXX was inserted into the EcoRI digested pBMNFlag-IRES-hCD2. pBMN-XLF-IRES-hCD2 and pBMN-FLAG-KU80-IRES-hCD2 and pBMN-FLAG-KU80ΔCTD-IRES-hCD2 were described previously[3,13].

**Western blot and antibodies.** Cell or tissue lysates were prepared in RIPA buffer (50 mM Tris-HCl pH 8.0, 150 mM NaCl, 0.1% SDS and 0.5% sodium deoxycholate, 1% NP40 and fresh proteinase inhibitor cocktail). Protein extracts were analysed by western blotting according to the standard protocols using primary antibodies

specific for PAXX (ab126353, Abcam, 1:1,000 dilution), XLF (A300-730A-1, Bethyl Laboratories, 1:1,000 dilution), Flag (M2, Sigma-Aldrich, 1:10,000 dilution), KAP1 (ab10484, Abcam, 1:1,000 dilution), phospho-KAP1 (ab70369, Abcam, 1:1,000 dilution). HRP-conjugated anti-rabbit and mouse secondary antibodies (GE Healthcare) were used and signal was detected using an ECL western blotting detection system (GE Healthcare). Uncropped images of all blots are available in Supplementary Fig. 6.

**Laser-induced recruitment of non-homologous end-joining factors.** Immortalized WT, $Ku80^{-/-}$, $Xlf^{-/-}$, $Paxx^{-/-}$ or $Xlf^{-/-}Paxx^{-/-}$ MEFs were seeded onto glass bottom 35 mm diameter plates at about $10^4$ cells per plate. Transient transfection of 1 μg GFP-Ku70 or 1 μg GFP-PAXX or 1 μg GFP-Lig4, respectively, with 1 μg DsRed-PCNA were conducted using Lipofectamine 2000 (ThermoFisher) following the manufacturer's instruction. To obtain reliable recruitment of GFP-Lig4 with the 405 nm laser, the transfected cells were sensitized with 10 μM of BrdU overnight before imaging. Live cell imaging combining laser micro-irradiation with confocal microscope was carried out using Nikon Ti Eclipse inverted microscope (Nikon, Inc.) equipped with A1 RMP (Nikon, Inc.) confocal microscope system (Nikon, Inc.) and Lu-N3 Laser Units (Nikon, Inc.). Laser micro-irradiation manipulation and time-lapse imaging was performed with the NIS Element High Content Analysis software (Nikon, Inc.), using a 405 nm laser with 100% energy level (output power on sample $\cong$ 110 μW, adequate for Ku70 accumulation). Relative intensity at laser-damaged sites was calculated as the mean value of the ratio of intensity of each micro-irradiation damaged sites to whole nucleus background. More than 10 individual cells were randomly chosen, tested and analysed for each data point. Mean value, standard deviation and outliers were calculated with Excel and GraphPad Prism online tools. For GFP-Ku70 recruitment, WT shows the average of 9 cells, $Xlf^{-/-}$ shows the average of 12 cells, $Paxx^{-/-}$ shows the average of 21 cells and $Xlf^{-/-}Paxx^{-/-}$ shows the average of 12 cells. For GFP-Lig4 recruitment, WT shows the average of 10 cells, $Xlf^{-/-}$ shows the average of 8 cells, $Paxx^{-/-}$ shows the average of 11 cells and $Xlf^{-/-}Paxx^{-/-}$ shows the average of 10 cells. In both GFP-Ku70 and GFP-lig4 recruitment, results of $Xlf^{-/-}$, $Paxx^{-/-}$ and $Xlf^{-/-}Paxx^{-/-}$ MEFs were compared with WT with two-way ANOVA to calculate the P value.

**Transient V(D)J recombination assay.** Extra-chromosomal V(D)J recombination assays were performed in v-abl transformed cell lines by transfection with JH200 (signal join) substrate plasmids as previously described.[3] V(D)J recombination efficiency was determined by the number of chloramphenicol–ampicillin double-resistant bacterial colonies among the total ampicillin-resistant colonies. The relative efficiency of SJ formation was calculated by setting the chloramphenicol–ampicillin double-resistant/total ampicillin-resistant colonies in the WT cells in each biological replicate as 100%.

**Data availability.** Vβ14 and Jβ sequence is available from the NCBI under Accession No. AE000665. The authors declare that all data supporting the findings of this study are available from the corresponding author upon request.

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

## Acknowledgements

We thank Dr Richard Baer for his comments and critical review of the manuscript. We thank Dr Andre Nussenzweig for providing the Ku80-deficient mice, from which we derived Ku80-deficient MEFs. We also thank Drs. Xiaochun Yu, Li Lan and David J. Chen for providing plasmids encoding tagged NHEJ factors used for live cell imaging. We also wish to thank Chen Li and Dr Chyuan-Sheng Lin for exceptional technical assistances in creating genetically modified mouse models. We apologize to colleagues, whose work could not be cited due to space limitations and was covered by reviews instead. This work is in part supported by NIH 5R01CA158073, 5R01CA184187, 1P01CA174653 and American Cancer Society Research Scholar Grant (RSG-13-038-01 DMC) to S.Z. S.Z. is the recipient of the Leukemia and Lymphoma Society Scholar Award. X.L. is a recipient of the Leukemia and Lymphoma Society Career Development Program Fellowship Award. W.J. was supported by NIH/NCI T32 training grant on Cancer Biology (NIH/NCI 5T32CA09503).

## Author contributions

X.L., Z.S., W.J. and B.J.L. conducted the experiments. X.L., Z.S. and S.Z. designed the experiments and analysed the data. X.L., Z.S. and S.Z. wrote the manuscript.

## Additional information

**Competing financial interests:** The authors declare no competing financial interests.

**How to cite this article**: Liu, X. *et al.* PAXX promotes KU accumulation at DNA breaks and is essential for end-joining in XLF-deficient mice. *Nat. Commun.* **8,** 13816 doi: 10.1038/ncomms13816 (2017).

**Publisher's note**: 

