## [Peer Review File · Nature Communications]

Editorial Note: Parts of this peer review file have been redacted as indicated to maintain the confidentiality of unpublished data. When text is deleted in rebuttals and referee reports, "[redacted]" has been added in that location.

Reviewers' comments:

Reviewer #1 (Remarks to the Author):

The article entitled "PAXX promotes Ku accumulation at DNA breaks and is essential for end-joining in XLF-deficient mice" by Liu et al reports on the characterization of a novel murine model of NHEJ deficiency, PAXX KO mice, in XLF-proficient and deficient background. PAXX, a recently discovered NHEJ factor, has been shown to promote NHEJ in human cells by promoting the accumulation of Ku at DSBs. These previous studies also demonstrated that PAXX is required for DSB repair and radioprotection and defined context-dependent nonoverlapping functions with XLF in human cells. The current article greatly expands our knowledge on PAXX functions in DNA repair by reporting for the first time on a murine model of PAXX deficiency. The newly generated PAXX KO mice show none of the phenotypes described in other NHEJ-deficient backgrounds. However, these phenotypes are elicited when bred to mice deficient for XLF, indicating non-redundant functions for the two factors. For example, double mutants are embryonic lethal and show extensive neural apoptosis similar to XRCC4- or LigIV-deficient mice. In addition, pro-B cells somatic KO for PAXX and XLF show striking defects in V(D)J recombination and double mutant pro-B and MEFs show genomic instability consistent with defective NHEJ at "general" DSBs. Mechanistically, the authors demonstrate that loss of PAXX results in reduced Ku at DSBs. While this finding is not novel, it indicates that the pathway is conserved between mice and humans and validates the murine model for additional studies.

Overall, the approaches used are the standard in the field, multiple biological and technical replicates are included, a large number of metaphases are analyzed in cytogenetic experiments and the statistical analyses are appropriate. The data is presented in a clear manner in the Results section and in the Figures and Tables. The conclusions are robust and are discussed appropriately in the context of the literature. Previous work is given credit.

Some minor questions/suggestions for improvement:

1. In Figure 2, the CD4/CD8 ratio appears to be reversed in all plots, are the axis labels switched?
2. In Figure 4, the spontaneous genomic instability in DKO MEFs is modest (Fig. 4C) and may not explain the severe growth defects in the same cells (Fig. 4A). XLF has been implicated in replication and XLF-deficient cells are hypersensitive to HU. Does loss of PAXX aggravate these phenotypes? What is the cell cycle distribution of these cells – for example, by PI staining or via BrdU incorporation.
3. As the authors point out, the current literature indicates that, in human cells, the mutual dependency of PAXX and XLF for DSB repair is context-dependent. Are DKO MEFs also hypersensitive to agents that introduce "cleaner" DSBs, such as etoposide or others?
4. In Figure 7E, there is still significant residual Ku at DSBs in the PAXX mutant, but no data is shown on PAXX/XLF double mutants, even though the cells are available to the authors. Does loss of XLF compromise Ku retention at DSBs further, or is the hypothesis that the more severe phenotypes in the DKO come from partial loss of Ku (due to PAXX deficiency) combined with partial loss of XRCC4 (due to loss of XLF)?

5. Grammar and spelling throughout the abstract and manuscript need careful editing. In the Introduction, the sentence in paragraph 3 that starts “. And the severe IR sensitivity...” is unclear.

Reviewer #2 (Remarks to the Author):

“PAXX promotes Ku-accumulation at DNA breaks and is essential for end—jointing in XLF-deficient mice” By Liu et al.

PAXX is the newest NHEJ core factor, which is structurally similar to XRCC4 and XLF. However, its animal model hasn't been generated and its physiological function is unknown. In this manuscript the authors studied the function of PAXX using the mouse model. They found that PAXX is dispensable for physiological NHEJ in wild-type mice, but required in XLF-/- mice. Mechanistically, PAXX is required for Ku-accumulation at DSB sites. Overall, it's an interesting finding that PAXX and XLF support NHEJ through distinct mechanisms.

Minor concerns:

- 1) Both ATM and DNA-PK are required for V(D)J recombination in XLF-/- pre-B cells. The authors discovered that the function of PAXX in NHEJ is distinct from ATM and XLF. Did they try whether PAXX functions together with DNA-PK?
- 2) Ku is required for the recruitments of all other NHEJ core factors, including Lig4. The authors found that PAXX is required for KU accumulation at DSB sites. Can the authors explain why the recruitment of Lig4 is not affected in PAXX-/- cells?

Reviewer #3 (Remarks to the Author):

Previous studies have indicated that the paralog of XRCC4 and XLF (PAXX, also called C9ORF142 or XLS) is a NHEJ factor. Studies in human cells indicated that PAXX is essential for NHEJ, and its loss results in hypersensitivity to various DNA damaging agents. However, its physiological role remains unclear. Here, Shan Zha and colleagues generated the PAXX-deficient mice, which surprisingly is not required for end-ligation measured by V(D)J recombination and class switch recombination assays. However, in combination with XLF-deficient animals, NHEJ is completely abrogated, leading to embryonic lethality associated with neuronal apoptosis, hyper-sensitivity to irradiation and defective chromosomal V(D)J recombination. Both both V(D)J associated signal- and coding-joint formation is compromised, similar to XRCC4 and LIG4 deficiency. These data clearly indicate that PAXX and XLF support NHEJ through distinct mechanisms. Their data suggest that PAXX facilitates the accumulation of KU at DNA ends and XLF enhances the recruitment of LIG4.

The experiments in this manuscript are well performed, convincing and provide a significant advance in the field.

The authors should address a few questions either by experimentation or clarification in the text:

- 1) They show that LIG4 recruitment is abrogated in XLF/PAXX MEFS. Is Ku recruitment also abrogated?
- 2) If PAXX facilitates KU accumulation and XLF enhances LIG4 accumulation, why does combined absence lead to lethality? The combined loss of LIG4 and KU results in mouse viability.
- 3) Xlf-/- cells, but not Paxx-/- cells require ATM/DNA-PK kinase. How does this fit with the idea that

XLF promotes LIG4 activity at DNA breaks, while PAX promotes KU accumulation at breaks. Is XLF recruitment of LIG4 dependent on ATM/DNA-PKcs?

4) RAG holds the DNA ends generated during V(D)J recombination in a post-synaptic complex, which allows end-ligation in the absence of XLF. Is RAG holding of ends linked to ATM/DNA-PKcs dependent recruitment of LIG4?

Point to Point Responses to the Review

The original comments from the reviewers were *italicized* for easy identification.

Reviewer #1 (Remarks to the Author):

The article entitled “PAXX promotes Ku accumulation at DNA breaks and is essential for end-joining in XLF-deficient mice” by Liu et al reports on the characterization of a novel murine model of NHEJ deficiency, PAXX KO mice, in XLF-proficient and deficient background. PAXX, a recently discovered NHEJ factor, has been shown to promote NHEJ in human cells by promoting the accumulation of Ku at DSBs. These previous studies also demonstrated that PAXX is required for DSB repair and radioprotection and defined context-dependent nonoverlapping functions with XLF in human cells. The current article greatly expands our knowledge on PAXX functions in DNA repair by reporting for the first time on a murine model of PAXX deficiency. The newly generated PAXX KO mice show none of the phenotypes described in other NHEJ-deficient backgrounds. However, these phenotypes are elicited when bred to mice deficient for XLF, indicating non-redundant functions for the two factors. For example, double mutants are embryonic lethal and show extensive neural apoptosis similar to XRCC4- or LigIV-deficient mice. In addition, pro-B cells somatic KO for PAXX and XLF show striking defects in V(D)J recombination and double mutant pro-B and MEFs show genomic instability consistent with defective NHEJ at “general” DSBs. Mechanistically, the authors demonstrate that loss of PAXX results in reduced Ku at DSBs. While this finding is not novel, it indicates that the pathway is conserved between mice and humans and validates the murine model for additional studies.

Overall, the approaches used are the standard in the field, multiple biological and technical replicates are included, a large number of metaphases are analyzed in cytogenetic experiments and the statistical analyses are appropriate. The data is presented in a clear manner in the Results section and in the Figures and Tables. The conclusions are robust and are discussed appropriately in the context of the literature. Previous work is given credit.

Some minor questions/suggestions for improvement:

1. In Figure 2, the CD4/CD8 ratio appears to be reversed in all plots, are the axis labels switched?

--- We apologized for this labeling error and thank the reviewer for pointing it out. We have now corrected it.

2. In Figure 4, the spontaneous genomic instability in DKO MEFs is modest (Fig. 4C) and may not explain the severe growth defects in the same cells (Fig. 4A). XLF has been implicated in replication and XLF-deficient cells are hypersensitive to HU. Does loss of PAXX aggravate these phenotypes? What is the cell cycle distribution of these cells – for example, by PI staining or via BrdU incorporation.

--- We thank the reviewer to bring this up. The proliferation as well as the cytogenetic assays in Fig.4 were performed in “primary” MEFs with normal G1/S (i.g. p53) and G2/M (i.g. ATM)

checkpoint function. Accordingly, we have now showed in Supplementary Figure 3A, in early passage (P1) *Xlf^{-/-}Paxx^{-/-}* MEFs, there is much less percentage of cells in S phase (marked by BrdU+) than in corresponding WT or single deficient MEFs. Therefore it is expected that only a fraction of cells with severe genomic instability could enter mitosis and be found in metaphase with breaks. In this case, metaphase based cytogenetic analyses provide a *conservative* estimation of chromosomal breaks in the primary cells. Consistent with this notion, the frequency of abnormal metaphase we reported for *Xlf^{-/-}Paxx^{-/-}* MEFs is comparable to similar measurement (by telomere FISH) in primary murine fibroblasts with defects in other end-joining deficient (*e.g.* DNA-PKcs/XLF DKO or KU KO) [1-4]. This data alone does not necessarily indicate a role for XLF/PAXX beyond NHEJ.

--- As suggested by the reviewer, we measured HU sensitivity in the primary MEFs. While cells deficient for either Paxx or Xlf alone display very moderate sensitivity to HU only at the higher doses, *Xlf^{-/-}Paxx^{-/-}* MEFs is not consistently sensitive to HU (Supplementary Figure 3C). HU preferentially targets replicating cells in S phase and *Xlf^{-/-}Paxx^{-/-}* primary MEFs have significant lower percentage of cells in S phase (Supplementary Figure 3A), which might contribute to the apparent lack of HU sensitivity in *Xlf^{-/-}Paxx^{-/-}* primary MEFs. Nevertheless, loss of Paxx does not significantly enhance the HU sensitivity in Xlf-deficient primary MEFs. These data are now included as Supplementary Figure 3C. Loss of XLF in human HCT116 cell has been linked to hypersensitivity to HU (Mol Cell Biol. 2015 Sep 1;35(17):3017-28). We note that level of HU sensitivity in *Xlf^{-/-}* primary MEFs is much less than what has been reported for HCT116 cells. Several factors might contribute to this difference, including but not limited 1) primary cells vs cancer cells, 2) HCT116 is deficient for MLH1, a member of the mismatch repair pathway that is responsible for correcting replication related errors, 3) human vs mouse cells.

3. As the authors point out, the current literature indicates that, in human cells, the mutual dependency of PAXX and XLF for DSB repair is context-dependent. Are DKO MEFs also hypersensitive to agents that introduce “cleaner” DSBs, such as etoposide or others?

- Following the reviewers' suggestion, we tested the sensitivity of *Xlf^{-/-}Paxx^{-/-}* MEF to etoposide (Supplementary Figure 3B). Both *Xlf^{-/-}* MEFs and *Paxx^{-/-}* MEFs display moderate yet significant hypersensitivity to etoposide, which is further exuberated in the *Xlf^{-/-}Paxx^{-/-}* MEFs. We note that etoposide traps Topo-isomerase II at the DNA ends and generates a “protein-blocked” DNA ends and prevents Ku loading [5]. Therefore etoposide generated ends require additional processing before it could be repaired by NHEJ, and etoposide sensitivity does NOT always correlate with end-joining defects. This additional need for end-processing might explain the context dependent sensitivity to etoposide in different cell lines (*e.g.* DT40 vs human cell lines). We have now included this data in Supplementary Fig. 3B.

4. In Figure 7E, there is still significant residual Ku at DSBs in the PAXX mutant, but no data is shown on PAXX/XLF double mutants, even though the cells are available to the authors. Does loss of XLF compromise Ku retention at DSBs further, or is the hypothesis that the more severe phenotypes in the DKO come from partial loss of Ku (due to PAXX deficiency) combined with partial loss of XRCC4 (due to loss of XLF)?

- We have now performed and quantified the recruitment kinetics of Ku in MEFs from all four genotypes (WT, Xlf^{-/-}, Paxx^{-/-}, Xlf^{-/-}Paxx^{-/-}) (New Figure 7E). The result suggests that Ku recruitment is NOT significantly affected in Xlf^{-/-} cells, and reduced significantly in both Paxx^{-/-} and Xlf^{-/-}Paxx^{-/-} cells. This result supports the conclusion that Paxx, but NOT Xlf, promotes efficient accumulation of Ku at the DNA ends. While both significantly lower than those in WT cells, Ku foci intensity is moderately yet consistently higher in Xlf^{-/-}Paxx^{-/-} cells than Paxx^{-/-} cells at 1min after irradiation. Given the ends can be efficiently ligated in Paxx^{-/-} cells, but not in Xlf^{-/-}Paxx^{-/-} cells, feedback accumulation of Ku due to persistent DNA double strand breaks might contribute to the higher intensity of Ku in Xlf^{-/-}Paxx^{-/-} cells than Paxx^{-/-} cells. We have now discussed this new data and its implication in the “Discussion” section, paragraph 2-3. Overall the data support the model that the severe end-ligation defects in Xlf^{-/-}Paxx^{-/-} cells is caused by partial reduction of Ku (due to Paxx deficiency) combined with partial loss of Lig4 (primarily due to Xlf deficiency) at the DNA ends.

5. Grammar and spelling throughout the abstract and manuscript need careful editing. In the Introduction, the sentence in paragraph 3 that starts “. And the severe IR sensitivity...” is unclear.

- We have carefully proof-read the revised manuscript and fixed the confusing sentences.

Reviewer #2 (Remarks to the Author):

“PAXX promotes Ku-accumulation at DNA breaks and is essential for end—jointing in XLF-deficient mice” By Liu et al.

PAXX is the newest NHEJ core factor, which is structurally similar to XRCC4 and XLF. However, its animal model hasn't been generated and its physiological function is unknown. In this manuscript the authors studied the function of PAXX using the mouse model. They found that PAXX is dispensable for physiological NHEJ in wild-type mice, but required in XLF^{-/-} mice. Mechanistically, PAXX is required for Ku-accumulation at DSB sites. Overall, it's an interesting finding that PAXX and XLF support NHEJ through distinct mechanisms.

Minor concerns:

1) Both ATM and DNA-PK are required for V(D)J recombination in XLF^{-/-} pre-B cells. The authors discovered that the function of PAXX in NHEJ is distinct from ATM and XLF. Did they try whether PAXX functions together with DNA-PK?

--- In Fig 7A, we showed, specific DNA-PKcs inhibitor (NU7441) effectively abolished end-ligation during V(D)J recombination in Xlf^{-/-} cells, but did NOT affect end-ligation in Paxx^{-/-} cells. We now discussed this result and its implications in the discussion sections.

2) Ku is required for the recruitments of all other NHEJ core factors, including Lig4. The authors found that PAXX is required for KU accumulation at DSB sites. Can the authors explain why the recruitment of Lig4 is not affected in PAXX^{-/-} cells?

- Thank this reviewer for bringing this up. We were puzzled by the similar question. So we performed and quantified the recruitment kinetics of Lig4 in MEFs from all four genotypes (WT, Xlf^{-/-}, Paxe^{-/-}, Xlf^{-/-}Paxe^{-/-}) using highly sensitive confocal microscope following BrdU sensitization (New Figure 7F, 10 μ M BrdU overnight incubation with 405nm laser, see method for details). Previous Lig4 recruitment was performed with a regular inverted Fluorescence microscope coupled with MicroBeam laser dissection system (365nm laser, not confocal). As showed before, Lig4 recruitment is dramatically reduced in Xlf^{-/-} deficient cells and complete absence in Xlf^{-/-}Paxe^{-/-} cells. Now using the more accurate co-focal microscope combined with BrdU sensitization, we were able to detect a very moderate yet significant reduction of Lig4 recruitment in Paxe^{-/-} cells, which is likely secondary to the Ku accumulation defects in Paxe^{-/-} cells. We now discuss this findings in the “Discussion” section, paragraph 2 - 3.

Reviewer #3 (Remarks to the Author):

Previous studies have indicated that the paralog of XRCC4 and XLF (PAXX, also called C9ORF142 or XLS) is a NHEJ factor. Studies in human cells indicated that PAXX is essential for NHEJ, and its loss results in hypersensitivity to various DNA damaging agents. However, its physiological role remains unclear. Here, Shan Zha and colleagues generated the PAXX-deficient mice, which surprisingly is not required for end-ligation measured by V(D)J recombination and class switch recombination assays. However, in combination with XLF-deficient animals, NHEJ is completely abrogated, leading to embryonic lethality associated with neuronal apoptosis, hyper-sensitivity to irradiation and defective chromosomal V(D)J recombination. Both both V(D)J associated signal- and coding-joint formation is compromised, similar to XRCC4 and LIG4 deficiency. These data clearly indicate that PAXX and XLF support NHEJ through distinct mechanisms. Their data suggest that PAXX facilitates the accumulation of KU at DNA ends and XLF enhances the recruitment of LIG4.

The experiments in this manuscript are well performed, convincing and provide a significant advance in the field.

The authors should address a few questions either by experimentation or clarification in the text:

1) They show that LIG4 recruitment is abrogated in XLF/PAXX MEFs. Is Ku recruitment also abrogated?

- In the **new Figure 7E**, we have now evaluated the recruitment of KU in MEFs from all four genotypes (WT, Xlf^{-/-}, Paxe^{-/-}, Xlf^{-/-}Paxe^{-/-}). The result indicates that Ku70 intensity at laser generated DNA breaks is significantly reduced in Xlf^{-/-}Paxe^{-/-} MEFs, but higher than those in Paxe^{-/-} MEFs. Please see response to reviewer 2-comment 4 for further discussion.

2) If PAXX facilitates KU accumulation and XLF enhances LIG4 accumulation, why does combined absence lead to lethality? The combined loss of LIG4 and KU results in mouse viability.

- We thank the reviewer for bringing this up. We were also intrigued by this apparent dilemma, until we compared the intensity of Ku70 in MEFs from all four genotypes. While Ku70 foci intensity is reduced in both *Paxx*^{-/-} and *Xlf*^{-/-}*Paxx*^{-/-} cells. The intensity of Ku is lower in *Paxx*^{-/-} cells than in the ligation defective *Xlf*^{-/-}*Paxx*^{-/-}, likely due to “continues” recruitment of Ku in the absence successful end-ligation. Ku deficiency rescued the embryonic lethality in *Lig4* deficient cells presumably by allowing end-resection and promoting alternative end-joining. Therefore the continue recruitment of Ku in *Xlf*^{-/-}*Paxx*^{-/-} cells likely block the end-ligation and also prevents alternative end-joining and leads to embryonic lethality. Consistent with this model, PAXX depends on KU for its initial recruitment to DNA breaks and likely modify the accumulation of KU without affecting initial recruitment of KU to the DNA ends. We have now further discussed this result and this point in the “Discussion” section, paragraph 2 – 3.

3) Xlf^{-/-} cells, but not *Paxx*^{-/-} cells require ATM/DNA-PK kinase. How does this fit with the idea that XLF promotes LIG4 activity at DNA breaks, while PAX promotes KU accumulation at breaks. Is XLF recruitment of LIG4 dependent on ATM/DNA-PKcs?

- We proposed that *Paxx* stabilizes Ku and potentially DNA-PKcs to promote the synapsis between two DNA ends. ATM/DNA-PKcs promotes end-synapses through the phosphorylation of chromatin bounded DNA damage response factors, such as H2AX and 53BP1. In this model, both ATM/DNA-PKcs as well as *Paxx* promote end-stability and synapsis. We have now clarified this in the “Discussion” section, paragraph 3. [REDACTED] We had examined *Lig4* recruitment in *Atm* and *Xlf* double deficient cells. These data were part of another manuscript that we are preparing. Although related, they were not included in this current study. We had not tested the impact of DNA-PK in *Lig4* recruitment in XLF-deficient cells.

4) RAG holds the DNA ends generated during V(D)J recombination in a post-synaptic complex, which allows end-ligation in the absence of XLF. Is RAG holding of ends linked to ATM/DNA-PKcs dependent recruitment of LIG4?

- This is an interesting question. Our current study addresses the functional interaction between XLF and PAXX in end- ligation. The requirement for either XLF or PAXX in efficient end-ligation is not only limited to RAG generated breaks, but also held true in post-mitotic neurons and in PPOI endonuclease generated breaks. Early this year, another study by Lescale. C., et al. on *Nature Communications* (2016 Feb 2;7: 10529) suggests RAG might function in a manner similar to ATM or its substrates in promoting end-joining in XLF-deficient cells [6]. However precise measurement for *Lig4* dynamics (within a minute) to RAG-dependent breaks have not been possible thus far. We now discussed this possible functional similarity between RAG, ATM and now potentially PAXX at different levels to promote end-ligation in *Xlf*-deficient cells.

Reference

1. Oksenyich, V., et al., *Functional redundancy between the XLF and DNA-PKcs DNA repair factors in V(D)J recombination and nonhomologous DNA end joining*. Proc Natl Acad Sci U S A, 2013. 110(6): p. 2234-9.
2. Jiang, W., et al., *Differential phosphorylation of DNA-PKcs regulates the interplay between end-processing and end-ligation during nonhomologous end-joining*. Mol Cell, 2015. 58(1): p. 172-85.
3. Ferguson, D.O., et al., *The interplay between nonhomologous end-joining and cell cycle checkpoint factors in development, genomic stability, and tumorigenesis*. Cold Spring Harb.Symp.Quant.Biol., 2000. 65: p. 395-403.
4. Ferguson, D.O., et al., *The nonhomologous end-joining pathway of DNA repair is required for genomic stability and the suppression of translocations*. Proc.Natl.Acad.Sci.U.S.A, 2000. 97(12): p. 6630-6633.
5. Aparicio, T., et al., *MRN, CtIP, and BRCA1 mediate repair of topoisomerase II-DNA adducts*. J Cell Biol, 2016. 212(4): p. 399-408.

REVIEWERS' COMMENTS:

Reviewer #1 (Remarks to the Author):

The revised manuscript "PAXX promotes KU accumulation at DNA breaks and is essential for end-joining in XLF-deficient mice" by Liu et al has fully addressed this reviewers' concerns by including novel data documenting the effect of PAXX deficiency on cell cycle; mutabst sensitivity to etoposide and HU; and Ku recruitment in PAXX/XLF double mutant cells.

Reviewer #2 (Remarks to the Author):

All my questions have been well addressed.

Reviewer #3 (Remarks to the Author):

The authors have done an excellent job in addressing all points from the reviewers. The MS should be published without delay.